# Trends in injury hospitalisations due to contact with snakes in Australia, 2002 to 2020: A registry data analysis for the Australian Venomous Injuries Project (AVIP)

**Afsana Afroz**●*, **Timothy N.W. Jackson**●, **Andrew D. Watt**●

Australian Venom Research Unit, Department of Biochemistry and Pharmacology, Faculty of Medicine, Dentistry and Health Sciences, University of Melbourne, Melbourne, Australia

* afsana.afroz@unimelb.edu.au

## Abstract

### Background

Snakebite envenoming is a significant yet often overlooked public health issue. Despite Australia's reputation for venomous snakes, hospitalisation rates for snake-bites have remained relatively low due to factors such as low population density, widely available antivenoms, and high-quality healthcare. However, population growth and climate change have increased snake sightings, raising concerns about shifts in snakebite incidence and seasonality. The Australian Venomous Injuries Project (AVIP) analysed hospitalisation data from 2002 to 2020 to assess trends and implications of snakebite-related injuries across Australia.

### Methods

Australian hospital separations data relating to contact with venomous snakes from the National Hospital Morbidity Database (NHMD), were analysed across the reference period 2002–2020. Data was provided by the Australian Institute of Health and Welfare (AIHW).

### Results

Between 2002 and 2020, a total of 10,763 hospitalisations with a principal ICD-10 diagnosis code of T63.0 'Toxic effect of contact with venomous animals, snake venom' or X20 'Contact with venomous snakes' were recorded across Australia, with an age-standardised rate of 2.6 (2.1-3.6) per 100,000 population. Males were hospitalised at more than twice the rate of females, and males aged 25–44 reported the highest number of bites. The Northern Territory had the highest age-standardised rate of hospitalisations from snakebite, with 7.6 (4.5-12.8) per 100,000 population, though hospitalisations in the Territory were stable across the study period. Hospitalisations

**Data availability statement:** The current investigation analysed Australian hospital separations data pertaining to contact with venomous snakes, as outlined in the methods section. This data belongs to the State Government Health Departments of Australia and is administered by the Australian Institute of Health and Welfare (AIHW). Our access agreement with the AIHW explicitly states that we are not to share the data in any form beyond the members listed on the agreement (being the coauthors listed on the manuscript). Separate approvals were also required to publish the data analysis in its current form. No special privileges were received with regard to accessing this data and other researchers may request access to this or other data through the AIHW (aihw.gov.au).

**Funding:** ADW and TNWJ received funding to undertake this work through a grant from the National Health and Medical Research Council of Australia (NHMRC; Grant# 13/093/002 AVRU; https://www.nhmrc.gov.au/). This grant provided salary costs for AA, TNWJ and ADW, and covered the associated research costs of the project work. The funders had no role in study design, data collection and analysis, decision to publish, or preparation of the manuscript.

**Competing interests:** The authors have declared that no competing interests exist.

in Tasmania, the Australian Capital Territory (ACT) and involving persons >65 years of age all increased by >5% across the study period.

## Conclusion

Despite population growth, urban expansion, and climate change, snakebite hospitalisation rates in Australia remained stable from 2002 to 2020. Mortality was not directly assessed but has remained stable in other reports. The observed increase in hospitalisations in Tasmania, the ACT, and older adults warrants further investigation to assess potential impacts on morbidity and mortality.

## Author summary

Snakebite is a neglected tropical disease with significant consequences to morbidity and mortality worldwide, particularly in impoverished regions. Venomous snakes are widespread across Australia, however, despite this, snakebite-related deaths remain rare in Australia, due to accessible and affordable antivenoms and high-quality public healthcare systems. This study analysed nearly two decades of Australian hospitalisation data (2002–2020) to understand national trends in snakebite injuries. During this time, there were >10,700 hospitalisations due to contact with venomous snakes, with rates remaining stable across the study period. Men were hospitalised more than twice as often as women, particularly those aged 25–44 years. The highest rates occurred in the Northern Territory and in remote regions, while hospitalisations increased among older adults and residents of Tasmania and the ACT. Brown snakes were responsible for nearly half of all envenoming cases, and around one in four envenomed patients received antivenom. These findings suggest that while the overall burden of snakebite in Australia has remained steady, emerging trends among older adults and in specific regions warrant attention. These findings may serve to inform public health strategies, clinical preparedness, and community education to further reduce the impact of snakebite injuries.

## Introduction

Snakebite, an often-underestimated cause of accidental death amongst the rural poor in tropical regions, is a significant global public health issue, especially in regions with high biodiversity and dense snake populations. Snakebite envenoming affects millions of people worldwide, posing a substantial mortality risk primarily in rural and agricultural communities of tropical and subtropical countries. According to [1], it is a major source of mortality globally, and it remains an important cause of preventable death. The WHO estimates that globally, there are approximately 5.4 million snakebites annually, resulting in 1.8–2.7 million cases of envenoming. These incidents lead to 81,410–137,880 deaths each year, with around three times as many individuals suffering from permanent disabilities and limb amputations [2].

Regions such as Sub-Saharan Africa, Southeast Asia, and South Asia experience the highest incidence of snakebites, with up to 200,000 cases of envenoming reported annually in Asia and 435,000–580,000 cases in Africa [2]. In South Asia, India experiences the highest mortality rate due to snakebite envenoming, with approximately 45,900 deaths each year. [3].

Despite being the only continent on which front-fanged venomous snakes outnumber non-venomous species, snakebite rates in Australia remain relatively low due in part to low population density in rural areas, with previous reports indicating that approximately 500–600 Australians are hospitalised each year following contact with venomous snakes [4,5]. Fatal snakebites are rare (2–4 deaths per year on average) [4–6] due to the wide availability of high-quality antivenoms across Australia's public health care system, with approximately 750 hospitals across the country maintaining a local supply [6,7]. When bites do occur, understanding contributing factors such as demographics, geographic locations (*e.g.*, living in rural or remote areas with close proximity to snake habitats), and the behavioural ecology of the snakes involved, can help refine our understanding of the impact of snakebite across the country. Equally, insufficient knowledge and lack of identification of venomous snakes, inappropriate first aid measures, and delays in reaching hospital or access to healthcare may exacerbate the consequences of snakebite.

In the early 1990s, it was estimated that approximately 3,000 people were bitten by snakes each year in Australia, with 200 of these individuals, at a minimum, receiving antivenom [8]. While the accuracy of these estimates was unclear, it had been established that venomous snakebites killed on average 1–4 people each year across the country [8,9]. More recent investigations have indicated that the number of people hospitalised in Australia following contact with venomous snakes is closer to 450–600 people each year [5,10,11]. Studies have reported that between 25–55% of patients with suspected bites from venomous snakes exhibit signs of clinical envenoming [5,6], with the lower rates likely including contact with non-venomous snakes, dry bites (bites from venomous species in which venom is not delivered/injected into the bitten individual), and "stick" bites (injuries caused by sharp sticks or twigs often in combination with the sighting of a nearby snake), in addition to clinically meaningful snakebites.

The Australian Venomous Injuries Project (AVIP) has previously investigated patterns of venomous injury between 2000–2013 using national incidence data from the Australian Institute of Health and Welfare (AIHW) [10]. Here, we build on these earlier investigations by analysing hospitalisation data due to snakebite collected between 2002–2020. The objectives of our analyses are to provide insights into the trends and implications of injury hospitalisations due to contact with venomous snakes in Australia. This registry data analysis aims to describe the patterns or trends of hospitalisations over the study period, examining variations across different demographics, geographic locations, and types of venomous snakes. It is hoped that this investigation will provide additional insights into the burden of snakebite envenoming across Australia and identify emerging trends to assist healthcare providers, policymakers, and public health officials in their efforts to mitigate the impact of snakebite on the Australian population.

## Methods

### Ethics

This project was approved by the University of Melbourne Human Research Ethics Committee (2022–12807), and the AIHW (2023–0009).

Australian hospital separations data provided by the Australian Institute of Health and Welfare (AIHW) data were examined to describe the incidence, trends and the characteristics of injuries arising from contact with venomous creatures resulting in hospital inpatient treatment.

### Data source

Case data were obtained from the National Hospital Morbidity Database (NHMD) for the period 2002–2020 through the AIHW, which includes all admitted episodes of care in Australian Hospitals. Records in the current analysis included all acute admissions and separations at public hospital in Australia relating to contact with a venomous snake as described below.

### Data definition

Separations by care type code 1.0 (acute admitted care) were used for analysis with other admission types (*e.g.*, 2.0: rehabilitation care, 8.0: other admitted care), and transfer separations were excluded from the database. The scope of the venomous injuries was defined by the International Statistical Classification of Diseases and Related Health Problems 10th Revision, Australian Modification (ICD-10-AM) codes relevant for each reference year [12].

### Data extraction

Data extracts were requested for principal diagnosis codes of X20 Contact with venomous snakes and lizards. Cases included and data were tabulated into T63.0 Toxic effect: snake venom or external cause codes X20 Contact with venomous snakes and lizards. Rates were age-standardised to the 2001 Australian population (per 100,000), using 6 age groups (0–4, 5–14, 15–24, 25–44, 45–64, 65+), with data on the remoteness area of usual residence defined by the Australian Bureau of Statistics' Australian Statistical Geography Standard Remoteness Structure 2011.

### Data definition

Data was defined following the external cause and diagnosis codes related to venomous animals as follows:

| Code | Description |
| --- | --- |
| X20 | Contact with venomous snakes and lizards |
| T63 | Toxic effect of contact with venomous animals |
| *T63.0* | *Snake venom* |

The X20 ICD code is an international code which accounts for countries where contact with venomous lizards is a medically significant event. Australia does not have indigenous medically significant venomous lizards, so this code has been shortened to X20 Contact with venomous snakes throughout the remainder of this article.

### Statistical analysis

The crude and age-standardised rates (per 100,000 population) of injury hospitalisations were calculated overall, as well as by age group and gender, state or territory of usual residence, and remoteness of usual residence. Additionally, the number and percentage of injury hospitalisations were presented overall and by gender for all sub-groups. Negative binomial regression models were used where the outcome variable was the natural log of the monthly number of hospitalisations and the primary independent covariate year. We report the outcomes of regression analyses as incident rate ratios (IRRs) with 95% confidence intervals (95% CIs). Data were prepared, analysed, and graphed using Microsoft Excel, Stata 17, and GraphPad Prism 10.2.3.

### Results

Between 2002 and 2020, a total of 10,763 hospitalisations with a principal ICD-10 diagnosis of T63.0 or X20 were reported across Australia. The overall crude hospitalisation rate was 2.6 (2.1-3.9) cases per 100,000 population, with an age-standardised rate of 2.6 (2.1-3.6) per 100,000 population (Table 1). The rate of male (3.5) hospitalisations were more than double that of female (1.6) hospitalisations and the trend was almost linear across this period. The range of this age-standardised rate for male across 2002–2020 was 2.8-4.9 and for female was 1.3-2.3.

The length of stay (LOS) for hospitalised cases due to contact with snakes was mostly short, with a mean of 1.3 days (±1.4 SD). Most cases (89.9%, 9,671 cases) were discharged on the same day of admission. An additional 9.3% (n = 1000) had an LOS of 2–7 days, while only 0.9% (n = 92) had an LOS exceeding one week, up to a month.

**Table 1. Crude and age-standardised rate (per 100,000 population) of injury hospitalisations due to contact with a venomous snake (X20), 2002 to 2020.**

|  | Male | Female | Persons |
|---|---|---|---|
| Number of cases | 7,340 | 3,423 | 10,763 |
| Crude rate/100,000 population | 3.5 | 1.6 | 2.6 |
| Age-standardised/100,000 population | 3.5 | 1.6 | 2.6 |

Notes:

1. Cases include those that have an external cause code of X20, Contact with Snakes in the record.

2. Age-standardised to the 2001 Australian population (per 100,000), using 6 age groups (0–4, 5–14, 15–24, 25–44, 45–64, 65+).

Source: AIHW National Hospital Morbidity Database.

## Injuries by age

People aged 25–44 accounted for approximately one third of all hospitalisations, followed by people aged 45–64 who accounted for more than a quarter of hospitalisations (Table 2 and Fig 1). Males aged 25–44 years had the highest number of hospitalisations (2,479), accounting for a third of male hospitalisations and more than 20% of overall hospitalisations. Males were hospitalised between 1.18-2.78 times more frequently than their age-matched female counterparts. Annual age-specific rates of venomous injury hospitalisations due to contact with snakes, by age and sex are provided in S1 Fig.

## Injuries by state and remoteness

The Northern Territory (NT) had the highest age-standardised rate of injury hospitalisations, with 7.6 (4.5-12.8) per 100,000 population (Table 3 and Fig 2), followed by Queensland (QLD; 4.3 (3.4-7.7) per 100,000), South Australia (SA; 2.9 (1.7-5.0) per 100,000), Western Australia (WA; 2.5 (1.7-3.7) per 100,000), and Tasmania (TAS; 2.5 (1.3-5.5) per 100,000). The Australian Capital Territory (ACT) recorded the lowest rate at 0.8 (0.2-2.5) per 100,000 population. Analysis of hospitalisations rates over time showed that venomous injury rates were largely stable over time, with the exception of the Northern Territory (NT) which had highest rates peak in 2011 (12.8 per 100,000) and 2017 (11.8 per 100,000; S2 Fig).

**Table 2. Number of hospitalisations due to contact with a venomous snake (X20), by age-group and sex, 2002 to 2020.**

| Age groups (years) | Male | | Female | | Person | |
|---|---|---|---|---|---|---|
|  | Number | % | Number | % | Number | % |
| 0–4 | 277 | 3.8 | 234 | 6.8 | 511 | 4.7 |
| 5–14 | 725 | 9.9 | 427 | 12.5 | 1,152 | 10.7 |
| 15–24 | 1,099 | 15.0 | 396 | 11.6 | 1,495 | 13.9 |
| 25–44 | 2,479 | 33.8 | 1,030 | 30.1 | 3,509 | 32.6 |
| 45–64 | 2,097 | 28.6 | 992 | 29.0 | 3,089 | 28.7 |
| 65+ | 663 | 9.0 | 344 | 10.0 | 1,007 | 9.4 |
| Total | 7,340 |  | 3,423 |  | 10,763 |  |

Notes

1. Cases include those that have an external cause code of X20, Contact with Snakes in the record.

2. Percentages may not sum to 100 due to rounding.

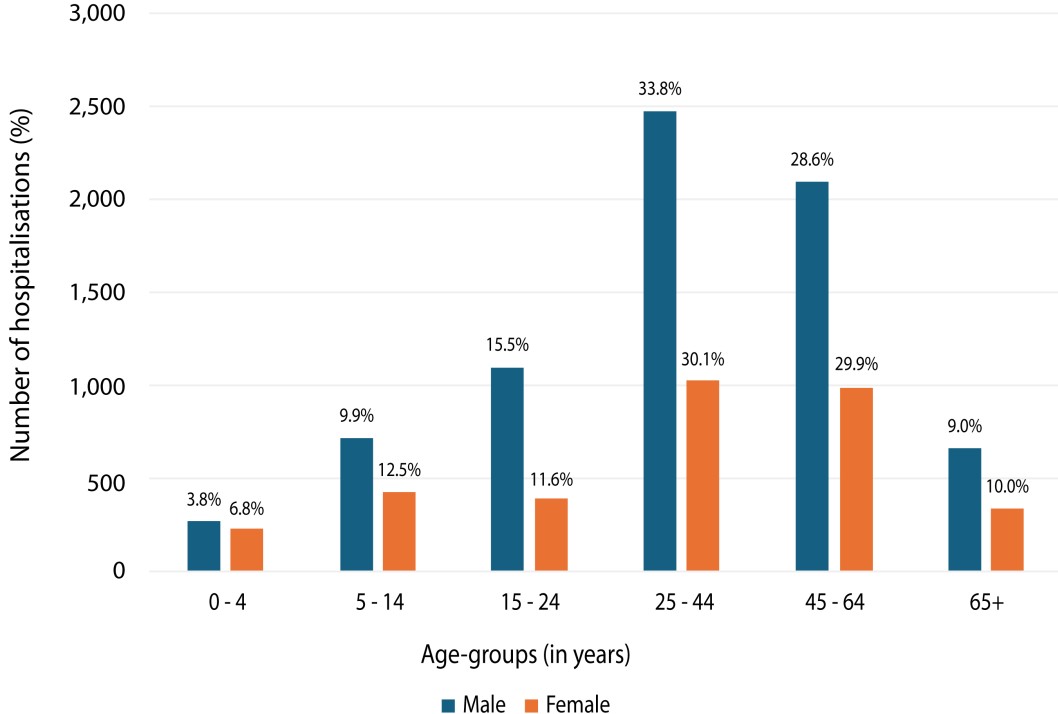

**Fig 1. Number of hospitalisations due to contact with a venomous snakes (X20), by age-group and sex, 2002 to 2020.** Notes 1. Cases include those that have an external cause code of X20, Contact with Snakes in the record.

**Table 3. Age-standardised rates of injury hospitalisations due to contact with a venomous snakes (X20), by state or territory of usual residence and remoteness of usual residence, 2002 to 2020.**

| State or territory of usual residence | Number | % |
|---|---|---|
| New South Wales | 2,992 | 27.8 |
| Victoria | 1,478 | 13.7 |
| Queensland | 3,554 | 33.0 |
| South Australia | 906 | 8.4 |
| Western Australia | 1,075 | 10.0 |
| Tasmania | 238 | 2.2 |
| Northern Territory | 334 | 3.1 |
| Australian Capital Territory | 45 | 0.4 |
| **Remoteness of usual residence** | | |
| Major Cities | 2,645 | 24.6 |
| Inner Regional | 3,848 | 35.8 |
| Outer Regional | 2,814 | 26.2 |
| Remote | 720 | 6.7 |
| Very Remote | 570 | 5.3 |

Notes:

1. Cases include those that have an external cause code of X20, Contact with Snakes in the record.

2. Age-standardised to the 2001 Australian population (per 100,000), using 6 age groups (0–4, 5–14, 15–24, 25–44, 45–64, 65+).

3. Based on the patient's state of usual residence.

4. Percentages may not sum to 100 due to rounding.

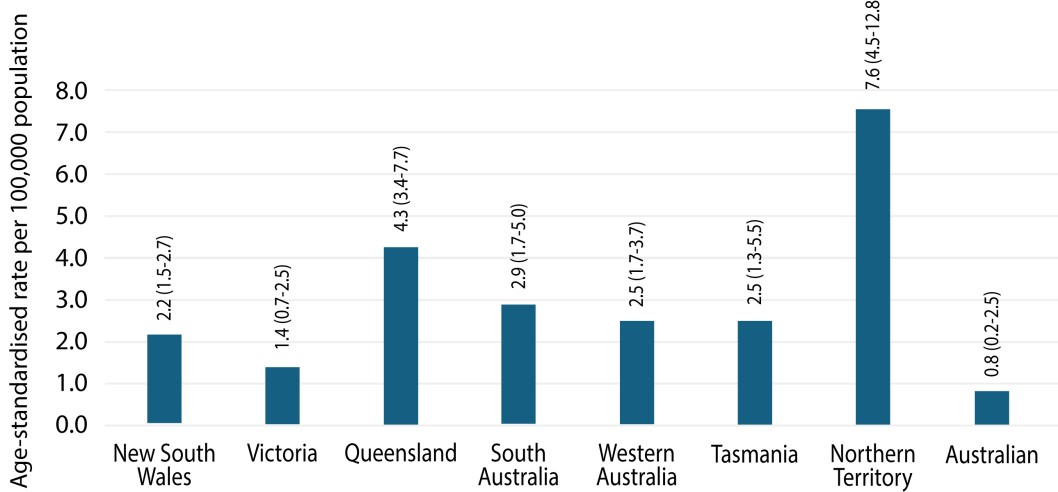

**Fig 2. Age- standardised rates (per 100,000 population) of selected injury due to contact with a venomous snakes (X20), by state or territory of usual residence, 2002 to 2020.** Notes 1. Cases include those that have an external cause code of X20, Contact with Snakes in the record. 2. Age-standardised to the 2001 Australian population (per 100,000), using 6 age groups (0–4, 5–14, 15–24, 25–44, 45–64, 65+). 3. Based on the patient's state of usual residence.

Age-standardised per-capita rates of hospitalisations increased with increasing remoteness, with residents of very remote areas experiencing a rate of 15.0 (5.9-24.3) per 100,000 population, nearly 15 times higher than residents in major cities (0.9 (0.7-1.1) per 100,000; Table 3 and Fig 3). The age-standardised hospitalisation rates across various geographical areas, categorised by remoteness of usual residence, remained relatively stable throughout the study period with an exception for the very remote areas which saw peak high rates in 2002 and 2022 (S3 Fig).

### Injuries by season, location and activity during injury

The number of injuries arising from contact with venomous snakes varied by season. In Australia, hospitalisations were highest during the summer months (December to February) and were almost five times higher during this time than hospitalisations recorded during winter (June to August). The pattern of injury hospitalisations began increasing from mid-Spring (October) and decreased by mid-Autumn (April; S4 Fig).

Of the total cases, only 7,286 (67.7%) cases listed information on the place of injury occurrence. Around 60% (4,193 cases) of these occurred in and around people's homes, females were more likely to be injured at home (67.6%) compared to males (52.3%). The place of injury occurrence was not specified for almost 1 in 4 of cases (23.3%; S5 Table).

The activity during the injury was also not specified for most cases, 5,889 (54.9%). Around one-fourths of hospitalisations due to contact with snakes occurred while working (24.1%) for income or engaged in other types of work followed by while engaging in leisure (4.5%) or sports (3.8%; S5).

### Injuries resulting in envenoming and/or requiring antivenom

Of the 10,763 snakebite cases recorded between 2002 and 2020, 42.1% (n = 4,529) were recorded as showing signs of clinical envenoming, (*i.e.*, 'T63.0 Toxic effect of contact with venomous snakes'; Table 4). Bites from brown snakes (*Pseudonaja spp.*) accounted for almost half of envenoming cases (44.3%, n = 2,008), followed by bites from tiger snakes (*Notechis spp.*; 16.25%, n = 736) and black snakes (*Pseudechis spp.*; 14.75%, n = 668).

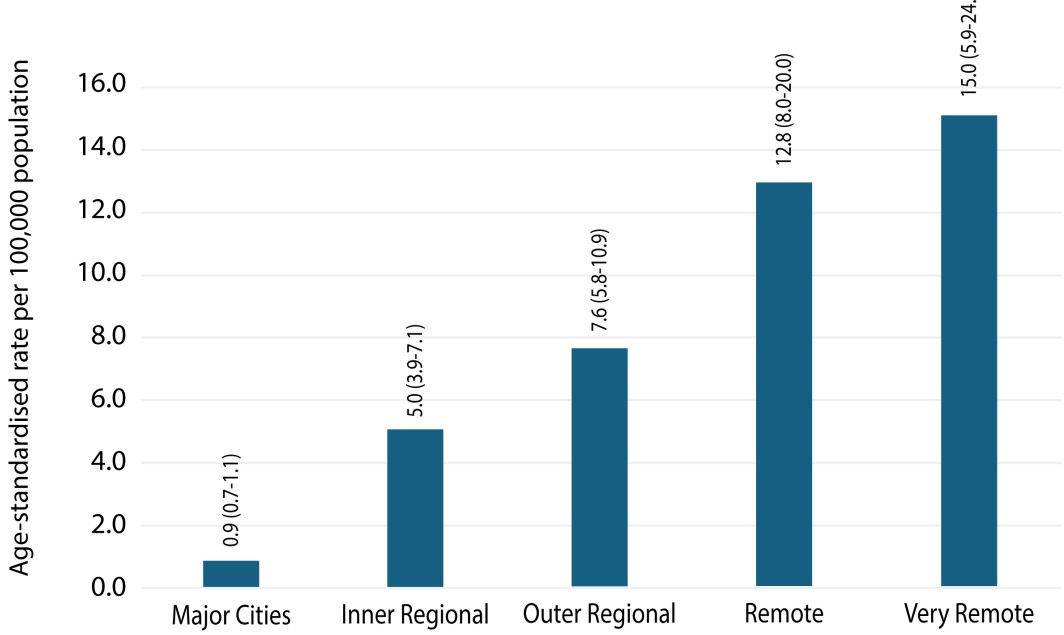

**Fig 3. Age- standardised rates (per 100,000 population) of injury due to contact with a venomous snakes (X20), by Standard Remoteness Structure, 2002 to 2020.** Notes 1. Cases include those that have an external cause code of X20, Contact with Snakes in the record. 2. Age-standardised to the 2001 Australian population (per 100,000), using 6 age groups (0–4, 5–14, 15–24, 25–44, 45–64, 65+). 3. Data on the remoteness area of usual residence are defined using the Australian Bureau of Statistics' Australian Statistical Geography Standard Remoteness Structure 2011.

**Table 4. Signs of clinical envenoming, (*i.e.*, 'T63.0 Toxic effect of contact with venomous snakes').**

| Type of snake | Cases (A) | | T63.0# (B) | | Antivenom administered (C) | % of all cases with antivenom administered (C/A) | % of cases with T63.0# and antivenom administered (C/B) |
|---|---|---|---|---|---|---|---|
| | n | % | n | % | | | |
| Brown snake | 4,416 | 42.91% | 2,008 | 44.34% | 455 | 10.3% | 22.7% |
| Black snake | 1,548 | 15.04% | 668 | 14.75% | 143 | 9.2% | 21.4% |
| Tiger snake | 1,187 | 11.53% | 736 | 16.25% | 324 | 27.3% | 44.0% |
| Taipan | 151 | 1.47% | 95 | 2.10% | 43 | 28.5% | 45.3% |
| Death adder | 279 | 2.71% | 151 | 3.33% | 41 | 14.7% | 27.2% |
| Sea snake | 106 | 1.03% | 55 | 1.21% | 4 | 3.8% | 7.3% |
| Other specified venomous snake | 301 | 2.92% | 127 | 2.80% | 15 | 5.0% | 11.8% |
| Unspecified venomous snake | 2,303 | 22.38% | 689 | 15.21% | 159 | 6.9% | 23.1% |
| **Total** | 10,291 | 100.00% | 4,529 | 100.00% | 1,184 | 11.5% | 26.1% |

#Total number of cases with **defined snake** and Principal diagnosis of 'Toxic effect of contact with venomous animals' (T63.0)

Antivenom was administered to approximately 1 in 4 patients with recorded signs of clinical envenoming (26.1%, n = 1,184), representing just over 1 in 10 of the total snakebite cases recorded. Antivenom administration rates were highest for patients bitten by taipans (*Oxyuranus spp.*), with almost half of patients with signs of envenoming receiving antivenom (45.3%, n = 43). Similar rates of administration were seen in patients with signs of envenoming following a tiger

snakes (*Notechis spp.*) bite (44.0%, n = 324). Bites from sea snakes (*Hydrophis spp.* and others) had the lowest percentage of cases receiving antivenom at 3.8% (n = 4). S6 Table further highlights the frequency and treatment patterns of different snake bite cases, emphasising the varying rates of antivenom administration among different types of snakes.

### Trends in venomous injuries due to contact with snakes

Regression analyses were conducted to assess whether associations existed between the monthly count of hospitalisations due to contact with snakes the year of injury which is summarised in Table 5 (for subgroups defined by age, sex, state or territory of usual residence and remoteness of usual residence). There were very few subgroups (highlighted by bolding) by patient demographic, state or territory of usual residence, and remoteness of usual residence, for which there was a statistically significant annual change in the monthly count of hospitalisation over the study period of 2002–2020.

Considering age groups, there was an annual increase in the monthly count (frequency) of cases among those aged between 45–64 years and 65 or older of 0.01 or 1% and 0.05 or 5%, respectively. That is, for each consecutive year of the study period, the mean monthly count of presentations in this group increased by 1% and 5%, respectively. South Australia, Tasmania, Northern territory and Australian capital territory had an annual increase in the monthly count between 2% and 6%. The major cities had an annual increase of 0.02%. Conversely, there was 1% - 3% decrease in the monthly

Table 5. Annual change in numbers of hospitalisations due to contact with venomous snakes (X20) for the study period of 2002 to 2020, by age group, sex, state or territory of usual residence and remoteness of usual residence: negative binomial regression analyses.

| Characteristic | Incident rate ratio (95% confidence interval), p-value | Trend |
| --- | --- | --- |
| **Age group (years)** | | |
| 0–4 | 0.97 (0.95–0.99), 0.014 | ↓ |
| 4–14 | 0.97 (0.95–0.99), 0.001 | ↓ |
| 15–24 | 0.99 (0.98–1.01), 0.653 | ↓ |
| 25–44 | 0.99 (0.98–1.01), 0.331 | ↓ |
| 45–64 | 1.01 (1.00–1.02), 0.002 | ↑ |
| 65 or older | 1.05 (0.94–1.07), 0.001 | ↑ |
| **Sex** | | |
| Male | 1.00 (0.99–1.01), 0.883 | – |
| Female | 1.00 (0.99–1.01), 0.735 | – |
| **State or territory of usual residence** | | |
| New South Wales | 0.99 (0.98-1.00), 0.808 | ↓ |
| Victoria | 0.99 (0.97-1.01), 0.191 | ↓ |
| Queensland | 1.00 (0.99-1.01), 0.998 | – |
| South Australia | 1.02 (0.99-1.04), 0.056 | ↑ |
| Western Australia | 0.99 (0.98-1.01), 0.434 | – |
| Tasmania | 1.05 (1.02-1.08), 0.001 | ↑ |
| Northern Territory | 1.02 (1.00-1.04), 0.065 | ↑ |
| Australian Capital Territory | 1.06 (1.01-1.12), 0.029 | ↑ |
| **Remoteness of usual residence** | | |
| Major Cities | 1.02 (1.01-1.03), 0.001 | ↑ |
| Inner Regional | 1.00 (0.99-1.01), 0.628 | – |
| Outer Regional | 0.99 (0.98-1.01), 0.023 | ↓ |
| Remote | 0.98 (0.96-1.00), 0.072 | ↓ |
| Very Remote | 0.98 (0.96-1.00), 0.061 | ↓ |

count (frequency) of cases among those aged between 0–44 years and there was 1% - 2% decrease in the monthly count (frequency) of cases among those reside in outer regional, remote, or very remote areas.

## Discussion

This study investigated the demographic and geographic trends of hospitalisation due to contact with snakes in Australia between 2002 and 2020 using data from the National Hospital Morbidity Database (NHMD) provided by the Australian Institute of Health and Welfare (AIHW). During this period, there were 10,763 hospitalisations overall, with a crude rate and an age-standardised rate of 2.6 cases per 100,000 population. 4,529 patients demonstrated clinical signs of envenoming and 1,184 of these cases received antivenom accounting for 33.5% of envenoming cases. Consistent with previous investigations, the rate of hospitalisation was higher among males than females particularly for individuals aged 15–44. The Northern Territory showed the highest rate of bites with 7.6 snakebites per 100,000 population (age-standardised) while the lowest rate was observed in the ACT with 0.8 per 100,000 population. Hospitalisation rates increased with the remoteness of residence, reaffirming the results of previous investigations [5,10]. Snakebite rates remained stable across NSW, VIC, QLD, SA, WA and NT, while TAS and ACT both showed statistically significant increase in injuries arising from snakebite across the study period.

In line with local and global studies, the number of hospitalisations in Australia due to contact with venomous snakes was higher for males than females, with males being hospitalised at rates 1.18 to 2.78 times higher than their age-matched female counterparts [5,6,8–10,13]. The greatest difference was observed in the 15–24-year age group. The highest hospitalisation figures were seen for patients aged 25–44, with this age group accounting for approximately a third of all patients. This was followed by individuals aged 45–64-years of age accounting for 28.7% of cases, consistent with earlier Australian studies [6,14]. Trend analysis indicated that significantly fewer young people (0–14 years) were seeking medical attention for venomous injuries due to contact with snakes while venomous injuries were increasing for individuals aged 45 years and over. Further investigations are suggested to better understand these shifting patterns in injuries and to determine trends in the years following COVID-19 lockdowns.

Around 42% of the snakebite cases recorded between 2002 and 2020 resulted in clinical envenoming. Brown snakes (*Pseudonaja spp.*) were the leading cause of envenoming accounting for 44.34% of cases, followed by tiger snakes (*Notechis spp.*; 16.25%), and black snakes (*Pseudechis spp.;* 14.75%). Antivenom was administered to a third of patients with signs of clinical envenoming with species specific rates ranging from 7.3% (for sea snakes) to 45.3% (for taipans). These figures are consistent with other reports that have found that envenoming rates from Australian elapids ranged from ~25% to 60% [5,6,11].

The level of antivenom administration reported in the current study was largely consistent with earlier studies investigating hospital level data [5], though lower than prospective studies which supplemented this data with additional clinical findings [6]. While this study was able to look at broad trends in antivenom usage it was unable to delineate whether the type or dosage of antivenom administered was suitable, or whether adverse reactions were in line with other reports (e.g., that hypersensitivity reactions to antivenom occurred in up to 25% of cases [14–16]). The variation of rates of both envenoming and antivenom administration noted here indicate that the current dataset may include contact with venomous snakes, dry bites (bites from venomous species in which venom was not delivered/injected into the bitten individual), and "stick" bites (injuries caused by sharp sticks or twigs often in combination with the sighting of a nearby snake), in addition to definitive cases of snakebite envenoming. More definitive rates can be garnered through prospective investigations such as the Australian Snakebite Project (ASP) [6] which collect additional clinical and supporting data; however, such investigations are resource intensive and a broader understanding of the impact of snakebite envenoming in Australia may be achieved when such findings are combined with population level analysis which provides an opportunity to assess national trends and identify areas needing additional attention and/or resources.

ASP hospital-based analyses similarly demonstrated that rural and remote presentations often involved delayed access to care and were associated with higher complication rates, reinforcing the geographic risk patterns observed in this study [15]. However, while rates in remote and very remote areas remained statistically unchanged, rates in outer regional areas significantly decreased and rates in major cities significantly increased across the study period. While statistically significant, it remains unclear whether the shifting trends across states and regions holds any clinical significance, or whether they are simply reflective of increasing population numbers in these areas. The number of injuries also varied by season, with the highest number of hospitalisations occurring during summer at a rate of almost six times higher than that observed in winter. Hospitalisations increased from the low of July through to the peak in January before decreasing once more. These findings were consistent with those of other studies investigating the global burden of snakebites [17] and with presumed peaks in reptile activity during in Spring-Summer in the southern regions of the country and during the build-up and wet season in the tropics. Further investigations are needed to better understand the factors driving these trends including the changing climate, shifts in population, and the impact of growing public awareness or other public health measures on these rates.

Earlier findings from our group and others [10] revealed that hospital admissions in Australia due to envenoming from contact with venomous snakes accounted for 18.6% of total envenoming-related hospital admissions. Previous reports indicate that hospital admissions due to the toxic effect of contact with snake venom (T63.0) were more than twice as likely to result in fatal outcomes compared to admissions involving venom from "other arthropods (T63.4)" [5,10]. ASP data corroborates this, documenting a consistently low annual case fatality rate of <1% despite frequent systemic envenoming, reflecting both timely healthcare access and effective antivenom therapy [6,18]. While this study did not report information related to fatal snakebites, previous studies have reported that between 2–7 fatal snakebites occur every year in Australia [5,10], suggesting that an estimated 38–133 fatal snakebites were expected over the 19 year study period. Given that most venomous injuries like snakebites, occur at or near a person's residence, particularly within major cities, these results highlight the importance of preventative measures and the administration of appropriate first aid (e.g., pressure immobilisation bandages) to minimise the risk of severe injury and death from negative encounters with Australian snakes.

Importantly, our hospitalisation-based registry approach can be complemented by emerging data-linkage cohorts such as the PAVLOVA study [19], which linked ambulance, ED, hospital, poison centre, toxicology unit and mortality data across NSW. PAVLOVA identified over 11,000 animal/plant envenoming exposures in a decade and demonstrated the added value of poison centre data for capturing substance-level details and repeated events. This linkage approach highlights that registry data such as those used by AVIP may underestimate the true burden of envenoming, particularly repeat presentations and outcomes beyond index hospitalisations.

Our work here revealed demographic, geographic, and species-specific variations in hospitalisations due to contact with venomous snakes from 2002 to 2020. While envenoming incidents due to contact with venomous snakes remain relatively infrequent in Australia, utilising data such as that provided here can be instrumental in helping Primary Health Networks align with the priorities set forth in the National Health Plan [20]. This could prove valuable for the identification of vulnerable population groups and areas and to support them with tailored clinical education, localised public health campaigns, and also in ensuring that antivenoms are distributed in accordance with local needs.

### Strengths and limitation

Although the AVIP dataset provides comprehensive national hospitalisation data, it does not capture the specific circumstances of each bite, such as the activity undertaken at the time of envenoming, environmental context, or occupational exposure. This limits direct interpretation for behavioural or situational prevention. However, the demographic, geographic, and temporal patterns identified in this study still provide a foundation for targeted prevention. For example, identifying high-incidence age groups, rural regions, and seasonal peaks can inform education and awareness campaigns, clinical preparedness, and first-aid training in at-risk communities. Integrating AVIP hospital data with prospective clinical datasets

**PLOS** **Neglected Tropical Diseases**

such as the Australian Snakebite Project (ASP) and data-linkage cohorts like PAVLOVA could enable more detailed understanding of bite circumstances and guide comprehensive prevention strategies.

A key strength of our study is the use of age-standardised rates, which underscores the importance of using consistent terminology and denominators to align priorities across various groups, from policymakers to health professionals. Our research, spanning over two decades, provides detailed insights into hospitalisations due to injuries from venomous snakes, categorised by demographics, geography, and type. However, there are limitations that should be acknowledged. The data provided by AIHW relies on ICD-10 coding, which may be subject to variations in coding quality and reporting accuracy. Additionally, our study does not address mortality data related to venomous bites from snakes. Furthermore, the lack of national standards for auditing the comparability of coded diagnoses and external cause data at local, state, or national levels hinders the ability to perform quantitative assessments of data quality on a national scale.

The lack of ancillary data to complement the hospital level data in the current investigations means that we were unable to investigate the rate and impact of pre-hospital treatments, such as the recommended pressure immobilisation bandage [21], the rate of complications, such as myotoxicity (ASP-23), thrombotic microangiopathy (ASP-30), and early cardiovascular collapse, as noted by ASP [22], nor the overall fatality rate of venomous bites across the study period. This is a significant limitation of the current hospitalisations data and the findings noted here are thus best utilised in concert with the studies described above (e.g., ASP; PAVLOVA).

It is also noted that the recorded identification of snake species in many of these cases are unlikely to be definitive, instead relying on some combination of clinical presentation, descriptions or photos provided by patients, knowledge of venomous regional species, consultation with state-based Poisons Information Centre, involvement of clinical toxicologists, and the use of snake venom detection kits (SVDKs), when and where available. Despite these limitations, the data here corroborated earlier findings from the Australian Snakebite Project (ASP) which included more definitive snake species identification methods and reported that brown snake envenoming accounted for 41% of venomous bites in a cohort of 1,548 Australian patients, with tiger snakes and black snakes accounting for 17% and 16% respectively [6].

### Conclusion

This study provides a comprehensive analysis of the demographic and geographic trends in hospitalisations due to contact with venomous snakes in Australia between 2002 and 2020. Significant variations were observed across different age groups, regions, and species of snake, highlighting the complexity of the factors contributing to envenoming incidents in Australia, as elsewhere in the world. However, the overall rate of hospitalisations remained steady across the study period. These findings underscore the importance of ongoing public health campaigns that continue to minimise the clinical impact of contact with venomous snakes. They also aim to provide some guidance on which factors driving these trends, such as regional differences, demographic shifts, environmental changes, and snake ecology may require additional insights to better mitigate the risks of venomous injuries. By identifying vulnerable populations and regions, this data may inform the development of targeted clinical education, localised public health campaigns, and the strategic distribution of antivenom.

### Supporting information

**S1 Fig. Age-specific rates of hospitalisations due to contact with a venomous snakes (X20), by age and sex, 2002–2020.**
(TIFF)

**S2 Fig. Year-wise trend of injury hospitalisations due to due to contact with a venomous snakes (X20), by state or territory of usual residence, 2002–2020.**
(TIFF)

**S3 Fig. Year-wise trend of injury hospitalisations due to due to contact with a venomous snakes (X20), by remoteness of usual residence, 2002–2020.**
(TIFF)

**S4 Fig. Number of selected injury hospitalisations due to due to contact with a venomous snakes (X20), by month and year, 2002–2020.**
(TIFF)

**S5 Table. Number of injury hospitalisations due to Contact with snakes, by place of injury and activity during injury, 2002–2020.**
(TIFF)

**S6 Table. The frequency and treatment patterns of different snake bite cases.**
(TIFF)

## Author contributions

**Conceptualization:** Afsana Afroz, Andrew D. Watt.

**Data curation:** Afsana Afroz, Andrew D. Watt.

**Formal analysis:** Afsana Afroz.

**Funding acquisition:** Timothy NW Jackson.

**Methodology:** Andrew D. Watt.

**Project administration:** Afsana Afroz.

**Supervision:** Andrew D. Watt.

**Validation:** Timothy NW Jackson, Andrew D. Watt.

**Visualization:** Afsana Afroz, Andrew D. Watt.

**Writing – original draft:** Afsana Afroz.

**Writing – review & editing:** Afsana Afroz, Timothy NW Jackson, Andrew D. Watt.

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
