## [Decision Letter · Decision Letter 0]

5 Jun 2025

Trends in injury hospitalisations due to contact with snakes and lizards in Australia, 2002 to 2020: A registry data analysis for the Australian Venomous Injuries Project (AVIP)

Dear Dr. Afroz,

Thank you for submitting your manuscript to PLOS Neglected Tropical Diseases. After careful consideration, we feel that it has merit but does not fully meet PLOS Neglected Tropical Diseases's publication criteria as it currently stands. Therefore, we invite you to submit a revised version of the manuscript that addresses the points raised during the review process.

Please submit your revised manuscript within 60 days . If you will need more time than this to complete your revisions, please reply to this message or contact the journal office at plosntds@plos.org. Please include the following items when submitting your revised manuscript:

We look forward to receiving your revised manuscript.

Kind regards,

Wayne Hodgson

Academic Editor

José María Gutiérrez

Section Editor

Shaden Kamhawi

co-Editor-in-Chief

Paul Brindley

co-Editor-in-Chief

**Journal Requirements:**

1) Please provide an Author Summary. This should appear in your manuscript between the Abstract (if applicable) and the Introduction, and should be 150-200 words long. The aim should be to make your findings accessible to a wide audience that includes both scientists and non-scientists. Sample summaries can be found on our website under Submission Guidelines:

3) Tables should not be uploaded as individual files. Please remove these files and include the Tables in your manuscript file as editable, cell-based objects. For more information about how to format tables, see our guidelines:

https://journals.plos.org/plosntds/s/tables 

4) We notice that your supplementary Figures, and Tables are included in the manuscript file. Please remove them and upload them with the file type 'Supporting Information'. Please ensure that each Supporting Information file has a legend listed in the manuscript after the references list.

**Reviewers' Comments:**

Reviewer's Responses to Questions

**Key Review Criteria Required for Acceptance?**

**Methods:**

-Are the objectives of the study clearly articulated with a clear testable hypothesis stated?

-Is the study design appropriate to address the stated objectives?

-Is the population clearly described and appropriate for the hypothesis being tested?

-Is the sample size sufficient to ensure adequate power to address the hypothesis being tested?

-Were correct statistical analysis used to support conclusions?

-Are there concerns about ethical or regulatory requirements being met?

Reviewer #1: The aims are described, although these are rather unattainable aims, which the paper ultimately doesn't address. The study design is limited by the source of the data which is routinely collected data from hospital, which can only provide general information on demographics, geographical differences and limited treatment data. The statistical analysis is described.

Reviewer #2: -Are the objectives of the study clearly articulated with a clear testable hypothesis stated? Objectives are not clearly mentioned

-Is the study design appropriate to address the stated objectives? To certain extend

-Is the population clearly described and appropriate for the hypothesis being tested? Yes

-Is the sample size sufficient to ensure adequate power to address the hypothesis being tested? Not applicable

-Were correct statistical analysis used to support conclusions? Yes

-Are there concerns about ethical or regulatory requirements being met? No

Reviewer #3: The Aim of the study is not clear and should be explicitly stated

The methods are too brief and very confusing to an international reader. Mtheods should be more detailed and divided into subtopics such as Data Source, Data Authenticity, Data Extraction, Definitions, Data analysis.

Information such as the identification of snake species (E.g. Brown snake etc.) should be stated in detail, how this identification is made in hospital settings and included in the database (E.g. specimen identification, venom detection kit etc?)

Methods should include definitions. Remove all definitions in the results and include them in the methods. E.g. “Envenoming was considered to have occurred when the principal diagnosis was listed as toxic effect from contact with venomous animals (T63.0).” currently in the results should be moved to methods.

**Results:**

-Does the analysis presented match the analysis plan?

-Are the results clearly and completely presented?

-Are the figures (Tables, Images) of sufficient quality for clarity?

Reviewer #1: The analysis is presented as per the Methods, but it is rather complicated in multiple tables and quite difficult to understand.

Reviewer #2: Does the analysis presented match the analysis plan? Yes

-Are the results clearly and completely presented? Yes, see specific comments below

-Are the figures (Tables, Images) of sufficient quality for clarity? Yes, minor modification needed

Reviewer #3: Figure 1 – Y axis label is incomplete. It should be “Number of hospitalisations”, X axis label is also incomplete - should be Age groups (in years)

Figures 2 and 3 are redundant because they duplicate the exact data presented in Table 3. Therefore, Fig 2 and 3 should be removed.

Table 3 – footnote “C.I. – Confidence Intervals” is confusing. Why it is stated there? The right side column head states that “Age-standardised rates per 100,000 population (min-max)”

**Conclusions:**

-Are the conclusions supported by the data presented?

-Are the limitations of analysis clearly described?

-Do the authors discuss how these data can be helpful to advance our understanding of the topic under study?

-Is public health relevance addressed?

Reviewer #1: Although the concluding paragraph suggests that the data supports the aims, in fact, these are just rather broad and ambiguous (motherhood type) statements. This paper will affect health policy - but how? Specifically how is this important in terms of improving the treatment of snakebite.

How will this data help with prevention. There is no information on the circumstances of the bite? This is key to preventing it. Is it an occupational hazard, or are they due to people interfering with snakes ? How many of the patients were snake handlers ? Previous studies have demonstrated that snake handlers make up to 10% of bites in Australia - please discuss, and refer to appropriate research from the Australian snakebite project.

There is no information on pre-hospital treatment, or patients that do not present to hospital. There is no information on deaths. This is not discussed as a limitation.

The identification of snakes is not defined - this is a clear limitation because it is well known that many people are bitten by "brown-coloured" snakes, which are not necessarily "brown snakes" - they must discuss this in reference to studies with definite snake identification.

Probably the largest weakness is that the authors provide a very poor literature review - there are 9 references, many from overseas. There have been numerous studies of snakebite from hospitals, poison centres, most from the Australian Snakebite project, which has not been discussed at all. Comparison of this data to the Australian snakebite project is essential, since it contains much more detailed information on all aspects of snakebite - demographics, circumstances, prehospital information, correct snake identification, clinical effects, outcomes and treatments.

The authors do not refer to the recent PAVLOVA study, a much more detailed and finer database linkage of poisoning and envenoming in Australia (for one State).

Cairns R, Noghrehchi F, Raubenheimer JE, Chitty KM, Isbister GK, Chiew AL, Brett J, Dawson AH, Brown JA, Buckley NA. Poisoning and envenomation linkage to evaluate outcomes and clinical variation in Australia (PAVLOVA): a longitudinal data-linkage cohort of acute poisonings, envenomations, and adverse drug reactions in New South Wales, Australia, 2011-2020. Clin Toxicol (Phila). 2024 Oct;62(10):615-624.

Reviewer #2: -Are the conclusions supported by the data presented? Yes

-Are the limitations of analysis clearly described? Yes

-Do the authors discuss how these data can be helpful to advance our understanding of the topic under study? Yes

-Is public health relevance addressed? Yes

Reviewer #3: Well-written

**Editorial and Data Presentation Modifications?**

Reviewer #1: Nil

Reviewer #2: (No Response)

Reviewer #3: Both terms, ‘envenoming’ (UK English) and ‘envenomation’ (US English) have been used in the manuscript. It would be appropriate to be consistent with one style.

**Summary and General Comments:**

Reviewer #1: The major weakness of this paper is that it is presented in isolation to any previous research done in Australia on snakebite. A simple PubMed search of "Australia" and "Snakebite" identifies hundreds of studies very quickly, and a search thru these excluding biomedical research, identifies 10s of papers by other research groups, clinical groups and poison centres on snakebite in Australia.

Reviewer #2: Overall, Australia has no native venomous lizards, and the details of lizards' injuries have made this manuscript less valuable, unless the lizard's injuries can be described in detail. No information about actual lizard injuries has been described. If possible, highly recommend excluding the lizard's details and presenting the data only on snakebites and envenomation. Especially under the results, presenting envenomation, antivenom treatments make no sense after including cases of non-venomous lizards.

Under discussion: Present incidence, envenomation, antivenom treatment for snakebites, state-wise, published in previous literature and compared with the newly extracted data.

Introduction: Include details of the incidence of snakebite and the envenomation in Australia, expressed per state.

Second paragraph, 3rd line: India is part of South Asia, not Southeast Asia.

Since most of the Australian territories are vast, it is important to present a map of the locations of the hospitals, indicating where the hot spots are. This is important to make the data more sense in policy making and implementing preventive strategies.

Discussion, paragraph 1: Why are these data not presented under the results, and only in the discussion?

Data present in many places in results and discussion: Please add the number where you have presented the data as a %.

Figure 1: Add % on the top of each bar.

S4: Delete the repeated word “due to” in the caption.

Many places: Unnecessary bold and italic formatting of: (Tables and Figures)

Reviewer #3: This manuscript describes the trends of hospitalisation due to snake and lizard -related injuries across Australia during 2002 to 2020. It concludes that despite population growth, urban expansion, and climate change, snakebite hospitalization rates in Australia remained stable during the above period. The manuscript includes some important data, for the global literature. However, there are some issues that need addressing.

PLOS authors have the option to publish the peer review history of their article (what does this mean? ). If published, this will include your full peer review and any attached files.

**Do you want your identity to be public for this peer review?** For information about this choice, including consent withdrawal, please see our Privacy Policy .

Reviewer #1: No

Reviewer #2: **Yes: ** Kalana Maduwage

Reviewer #3: No

**Figure resubmission:**

**Reproducibility:**



---

## [Editor Report · Decision Letter 1]

17 Nov 2025

Dear Sr Afroz,

We are pleased to inform you that your manuscript 'Trends in injury hospitalisations due to contact with snakes in Australia, 2002 to 2020: A registry data analysis for the Australian Venomous Injuries Project (AVIP)' has been provisionally accepted for publication in PLOS Neglected Tropical Diseases.

Best regards,

Wayne Hodgson

Academic Editor

José María Gutiérrez

Section Editor

Shaden Kamhawi

co-Editor-in-Chief

Paul Brindley

co-Editor-in-Chief

The authors have adequately responded to the concerns of the reviewers.

---

## [Editor Report · Acceptance letter]

Dear Dr Afroz,

We are delighted to inform you that your manuscript, "Trends in injury hospitalisations due to contact with snakes in Australia, 2002 to 2020: A registry data analysis for the Australian Venomous Injuries Project (AVIP)," has been formally accepted for publication in PLOS Neglected Tropical Diseases.

Best regards,

Shaden Kamhawi

co-Editor-in-Chief

Paul Brindley

co-Editor-in-Chief
